# Fatty Acids as Nutritional Therapy for NAFLD: A Bibliometric Analysis of Research Trends and Future Directions

**DOI:** 10.3390/foods14244277

**Published:** 2025-12-12

**Authors:** Zicheng Huang, Xiangjun Zhan, Jun Jin, Xingguo Wang, Qingzhe Jin

**Affiliations:** 1State Key Laboratory of Food Science and Resources, School of Food Science and Technology, Jiangnan University, Wuxi 214122, China; 2Food Laboratory of Zhongyuan, Luohe 462300, China; 3Department of Food Science and Technology, School of Agriculture and Biology, Shanghai Jiao Tong University, Shanghai 200240, China

**Keywords:** fatty acids, non-alcoholic fatty liver disease, bibliometrics, ω-3 fatty acids, ω-6 fatty acids, Torreya grandis oil

## Abstract

The global prevalence of non-alcoholic fatty liver disease (NAFLD) is 25%, and its onset is closely related to fatty acid metabolism disorders. With the rise of the concept of non-drug treatment, the intervention potential of unsaturated fatty acids (especially ω-3/ω-6 fatty acids) has become a research hotspot, but the field’s development trend has not been systematically evaluated. Based on bibliometric analysis, 4509 NAFLD fatty acid-related articles in the Web of Science core collection were retrieved, and CiteSpace and VOSviewer were used to analyze the country, institution, and author cooperation networks, and keyword evolution. The annual publication volume peaked in 2022 (316 articles). China led the research output, but the United States had a significant lead in influence. The key author cluster was centered on Sanyal Arun (USA) and Li Y (China); the University of California system and the French National Institute of Health were high-impact institutions. The research topic has shifted from pathological mechanisms (“insulin resistance” and “oxidative stress”) to clinical intervention (“ω-3 fatty acids” and “double-blind trials”). The research on fatty acids in NAFLD has shifted from a stable period to a transitional period. The key words “ω-3 fatty acids”, “double-blind trials”, and “short-chain fatty acids” indicate that nutritional intervention has entered the evidence-based verification stage. Future research should explore the therapeutic potential of unsaturated fatty acids (e.g., ω-6/ω-9 fatty acids) and specialty oils, such as *Torreya grandis* oil, as novel dietary interventions.

## 1. Introduction

Global affluence has fueled non-alcoholic fatty liver disease (NAFLD), which now affects about 25% of adults (29.6% in Asia, 31.8% in the Middle East, and 13.5% in Africa) [1,2,3]. Hepatic lipid oversupply or impaired clearance initiates steatosis, which may progress via chronic inflammation to non-alcoholic steatohepatitis (NASH), fibrosis, and cirrhosis [3,4]. Although liver biopsy remains the gold standard to distinguish simple steatosis from NASH, non-invasive imaging techniques such as computed tomography (CT) and ultrasound are more commonly employed in clinical practice [5]. Recent advances reveal key mechanisms beyond insulin resistance in NAFLD pathogenesis. Hepatic lipid dysregulation involves SREBP-1c-driven de novo lipogenesis and impaired PPARα-mediated fatty acid oxidation [6]. The gut–liver axis plays a critical role, with microbial metabolites (SCFAs, bile acids) modulating host metabolism [7]. Genetically, PNPLA3-I148M remains a key risk factor, while emerging targets like NNMT regulate NAD+ metabolism and mitochondrial function [8]. Concurrently, TLR4/NLRP3 inflammasome activation and immune cell metabolic reprogramming (e.g., TH17 cells) drive NASH inflammation and fibrosis [9,10]. These interconnected mechanisms provide new frameworks for therapeutic intervention. Current evidence-based management emphasizes lifestyle modification and balanced, low-fat dietary patterns; however, poor adherence has propelled nutrition research into specific fatty acids as therapeutic targets.

Emerging evidence suggests that fatty acid balance is involved in NAFLD pathogenesis. Elevated serum free fatty acids (FFAs) correlate with hepatocellular injury markers [11] while hepatic long-chain polyunsaturated fatty acids (PUFAs) are depleted in NASH, concomitant with upregulation of *FADS1*, *FADS2,* and *PNPLA3* expression [12]. Furthermore, a large-scale UK Biobank analysis revealed that habitual PUFA intake inversely associates with NAFLD incidence [13], underscoring the potential of unsaturated fatty acids.

The ω-3/ω-6 fatty acid family within unsaturated fatty acids has garnered particular attention in relation to NAFLD. A study by Sertoglu et al. linked a high ω-6/ω-3 ratio to hepatic steatosis, noting that hepatic ω-6 content in NAFLD patients correlates with disease severity [14]. Shama et al. found that an elevated ω-6/ω-3 ratio may contribute to liver steatosis and inflammatory responses, thereby exacerbating NAFLD progression [15]. A meta-analysis by Yu et al [16]. confirmed that ω-3 supplementation reduced serum alanine aminotransferase (ALT) levels in NAFLD patients [MD = −9.18, 95% CI (−12.41, −5.96), *p* < 0.00001], indicating benefits for liver function. Current evidence suggests a strong association between ω-6 fatty acids and the onset and progression of NAFLD, prompting research into specific ω-6 fatty acids, such as delta-5 fatty acids and *Torreya grandis* oil/*Torreya grandis* seed oil.

In recent years, although studies on NAFLD and fatty acid metabolism have expanded substantially [17,18], the field still lacks comprehensive quantitative analyses of its global structure, collaborative networks, historical development, and future trends. Bibliometrics—an interdisciplinary method that applies mathematical and statistical approaches—enables the quantification of research status and trends by analyzing authorship patterns, institutional and national collaborations, and keyword hotspots. Here, this approach was used to chart the translational trajectory of fatty acids as therapeutic agents in NAFLD. This trajectory can be understood by evaluating research fronts through the lens of evidence maturity and clinical applicability. To address this gap, the present study employs bibliometric analysis to investigate NAFLD and fatty acid metabolism research, focusing on high-quality literature published over the past 25 years.

## 2. Materials and Methods

### 2.1. Data Sources

This study retrieved literature from the Science Citation Index Expanded (SCI-E) database, which is part of the Web of Science (WOS) Core Collection. Renowned for its comprehensive coverage, rigorous indexing standards, and high-quality retrospective academic records, the WOS Core Collection serves as a gold standard data source in bibliometric research, ensuring the reliability and academic validity of the analytical results.

### 2.2. Search Strategy

The search strategy was formulated to comprehensively capture relevant literature by combining key terms related to the research subject. The specific search formula was as follows: TS = ((“Nonalcoholic fatty liver disease” OR “NAFLD”) AND (“fatty acid”)) OR ((“Omega-6 fatty acid” OR “n-6 fatty acid”) AND (“Nonalcoholic fatty liver disease” OR “NAFLD”)). To ensure the academic relevance and quality of the included literature, the search was restricted to documents categorized as articles and review articles, that are written in English, and were published between 1 January 2001 and 11 June 2025. All searches were executed on 11 June 2025 to mitigate any potential biases arising from database updates during the extended search and data collection period. Literature screening and data extraction were performed independently by two researchers to ensure accuracy and minimize bias. This involved (a) title/abstract screening using inclusion criteria and (b) full-text assessment of eligible studies. All discrepancies were resolved through consensus, with a third senior researcher arbitrating unresolved disagreements. This protocol ensured reliable and reproducible record selection.

### 2.3. Analysis Tools

CiteSpace (version 6.2.4), a widely used bibliometric analysis software for exploring knowledge networks and academic evolution [19,20], was employed to process and analyze the retrieved WOS literature data. By configuring appropriate threshold parameters, this tool effectively revealed the knowledge structure, research frontiers, and evolutionary trends in the field. CiteSpace facilitated the construction of networks for author, institution, and country/institution collaboration, as well as keyword co-occurrence maps. The key parameters were set as follows: years per slice: 1 year; node type: single independent analysis; pruning algorithm: Pathfinder; visualization: cluster view—static and merged network display.

VOSviewer (version 1.6.16), developed by the Center for Science and Technology Studies (CWTS) at Leiden University in the Netherlands [21,22], was employed for supplementary visualization analysis. VOSviewer facilitated the depiction of the temporal evolution of authors, institutions, and countries/regions, as well as the spatial distribution of research hotspots. All retrieval and analysis procedures were performed independently by the authors to ensure methodological consistency.

### 2.4. Statistical Methods

This study employed a bibliometric approach, relying exclusively on descriptive statistical indicators (e.g., publication counts, citation frequencies, and collaboration networks) to characterize research activity. No inferential or hypothesis-driven statistical analyses were applied.

## 3. Results

### 3.1. Publication Trends and Output Analysis

A total of 4509 articles and reviews were incorporated into this study. Figure 1 illustrates the trajectory of published articles and citations from 2001 to 2025. The data indicate that from 2001 to 2004, the field was considered to be in its nascent stage, with a limited number of high-quality studies published. Between 2005 and 2012, a gradual increase in the number of studies was observed and a phase of rapid growth was observed in the field, reaching its peak in 2022. The subsequent decline and stabilization signal a transition into a consolidation phase, where research efforts are likely shifting from volume-driven expansion to resolving complex mechanistic and translational questions.

### 3.2. Authorship and Collaboration Patterns

A bibliometric visualization analysis of the authors in this field was conducted, identifying a total of 21,089 contributors. The author collaboration network exhibits a loosely connected structure with low centrality, indicating the absence of a tightly knit, dominant core of researchers in the field. The analysis of the author collaboration network diagram (Figure 2A) revealed frequent collaborations between Li Y, Zhang J, and Wang H, while a close partnership was observed between Hodson Leanne and Yki Jarvinen Hannele. The author hotspot region map (Figure 2B) showed concentrated activity around Sanyal Arun, Li Y, and Hodson Leanne, suggesting that these authors are from two primary research clusters in this domain. The authors with the highest research impact based on explosive growth trends included Hodson, Leanne (2019–2022), Zhang, Li (2019–2020), and Chen, Wei (2020–2023) (Figure 3). Table 1 lists the top ten authors ranked by publication volume, with Li Y (71 publications), Zhang Y (57 publications), and Li J (56 publications) leading the count. In terms of total citations, Li Y (2090 citations), Liu Y (1623 citations), and Wang H (1461 citations) were identified as the most highly cited authors. Notably, eight of the top ten authors were found to be from Chinese institutions.

A comprehensive visualization of the global cooperation network among countries and regions in this field was generated. The analysis identified 88 participating countries and regions. The cooperation network diagram, depicted in Figure 4A, revealed that China had the highest number of nodes and publications, followed by the United States. However, the United States was found to have more cooperative links with other countries compared to China, indicating a stronger tendency toward international collaboration, particularly with Italy and England. The countries and regions demonstrating the most significant growth in collaborative activity were the United States (2001–2009), Italy (2001–2012), and Japan (2005–2012), as illustrated in Figure 4B. Table 2 presents the top ten countries and regions ranked by publication volume. Among these, China (1690), the United States (1031), and South Korea (302) were ranked highest in publication output, while the United States (87,313 citations), China (47,648 citations), and Italy (26,417 citations) led in total citation count. The United States held the highest centrality (0.69), identifying it as the central hub for international collaboration, followed by China (0.40), Italy (0.14), South Korea (0.11), and England (0.11). From a global geographical perspective, these countries are located in Asia, five in Europe, and two in America. Among the top ten countries, only China is classified as a developing country, while the remaining countries are developed.

### 3.3. Institutional Contributions and Networks

A comprehensive bibliometric analysis of the institutions involved in this field was conducted, identifying 88 participating institutions. The cooperation network diagram was examined (Figure 5A), and the analysis of node sizes and interconnectivity indicated a tripartite relationship among the Ministry of Education of China, the University of California system, and the U.S. Department of Veterans Affairs. The University of California system was found to have a significantly higher number of collaborative links, while the Ministry of Education of China was identified as the largest node in the cooperation network. With the highest centrality score (0.34), the University of California System served as the pivotal hub for institutional collaboration, significantly leading other key institutions such as the Ministry of Education China (0.16) and the US Department of Veterans Affairs (0.13). The institutions demonstrating the highest levels of research dynamism included the U.S. Veterans Health Administration (VHA), the U.S. Department of Veterans Affairs, Peking Union Medical College, the University of Gothenburg, and the Shanghai University of Traditional Chinese Medicine (Figure 5B). Table 3 presents the top ten institutions ranked by the number of publications. The top three were the Ministry of Education China (166 articles), the Chinese Academy of Sciences (108 articles), and the University of California system (102 articles). Regarding total citations, the University of California system (9583 citations), the National Center for Biomedical Research (8392 citations), and the U.S. Department of Veterans Affairs (8158 citations) ranked highest.

### 3.4. Journal Distribution and Impact

The top ten journals by article count were analyzed in this study (Table 4). The top three journals by publication volume were *Nutrients, International Journal of Molecular Sciences*, and *Hepatology*. According to the latest impact factors (IFs), the top three journals were *Journal of Hepatology* (IF = 26.8), *Hepatology* (IF = 13), and *Food & Function* (IF = 5.1). *Hepatology* was found to have the highest average number of citations per article (174.85 times), significantly exceeding all others, indicating broad recognition of its research quality within the academic community. In summary, the CiteScore rankings of these 10 journals ranged from Q1 to Q4, with most journals concentrated in the fields of nutrition (n = 3), hepatology (n = 3), and molecular biology (n = 1). Two of the remaining journals, *Scientific Reports* and *Plos One*, are considered multidisciplinary platforms for the natural sciences. *Frontiers in Pharmacology* is a specialized journal in the field of pharmacology. These ranking may serve as a reference for researchers considering submission targets.

### 3.5. Highly Cited Literature in the Field

The ten most frequently cited high-quality publications in this field, primarily spanning 2001 to 2022, are presented in Table 5. These works primarily address topics such as metabolic syndrome and molecular mechanisms. Overall, the literature was found to consist largely of mechanistic reviews and fundamental research, indicating that research on NAFLD has largely focused on exploring pathological mechanisms, with comparatively less attention given to clinical translational studies. Specifically, the distribution of articles consisted of five basic research articles (experimental/animal models), four systematic reviews, and one clinical research study. These studies were primarily published in leading journals in the fields of liver disease and metabolism, highlighting the interdisciplinary nature of research conducted at the intersection of these domains.

### 3.6. Keyword Evolution and Research Hotspots

The most frequently used keywords in this domain were identified to be “insulin resistance” (906 occurrences), “hepatic steatosis” (892 occurrences), “fatty liver” (691 occurrences), “oxidative stress” (673 occurrences), “expression” (652 occurrences), “non-alcoholic fatty liver disease” (641 occurrences), “steatohepatitis” (634 occurrences), “obesity” (554 occurrences), “non-alcoholic steatohepatitis” (553 occurrences), and “non-alcoholic fatty liver disease” (548 occurrences), as shown in Figure 6A. These keywords highlight the central focus and evolutionary trajectory of research concerning NAFLD. Figure 7 illustrates the chronological progression of keywords within this field. The keywords with the highest centrality were hepatic steatosis (0.14), nonalcoholic fatty liver disease (0.14), and oxidative stress (0.12), identifying them as the most pivotal conceptual hubs within the research landscape. The initial phase was focused on investigating the mechanisms of oxidative stress and gene regulation in the progression from hepatic steatosis to steatohepatitis, as well as the role of fatty acids in dietary interventions for NAFLD. In the subsequent phase, keywords such as “short-chain fatty acids”, “omega-3 fatty acids”, “management”, and “double-blind” emerged, indicating a growing scholarly interest in both the disease and its clinical applications. The top five keywords demonstrating strong citation bursts were “metabolic syndrome” (2006–2015), “insulin resistance” (2005–2010), “steatohepatitis” (2004–2013), “adipose tissue” (2001–2016), and “gene expression” (2009–2015), as illustrated in Figure 6B. These findings indicate that these keywords represented active areas of research during the respective periods, suggesting that future research hotspots may include topics such as “protecting ppar”, “γ”, “double-blind”, and “short-chain fatty acids”.

## 4. Discussion

The bibliometric trends presented in this study do not merely quantify research activity; they collectively chart the translational pathway of fatty acids from molecular targets to therapeutic candidates in NAFLD. Our analysis reveals that this pathway was not uniform and was defined by distinct phases of evidence maturity and conceptual evolution. This first bibliometric mapping of fatty-acid-centered NAFLD/MAFLD literature (Scopus, 2001–2023) shows a rapid growth phase from 2005 to 2012, peaking in 2022, followed by a plateau that coincided with the 2020 redefinition of MAFLD, which redirected the research focus toward metabolic-syndrome-driven fibrosis [24,33]. China led in publication output (1690 articles, 37.5%), yet lagged behind in citation impact (47,648 total citations vs. 87,313 for the United States; H-index: 93 vs. 147). Although eight of the ten most prolific authors were from Chinese institutions, China’s international collaboration remained limited compared to the United States, which demonstrated a dense cooperative network with the European Union, the United Kingdom, and Japan. Strengthening multinational partnerships—particularly with high-impact U.S. consortia such as the University of California system, the Department of Veterans Affairs, and the National Institutes of Health—may enhance translational research depth.

High-impact mechanistic studies were predominantly published in *Hepatology* (IF = 13) and the *Journal of Hepatology* (IF = 26.8), while intervention trials were mainly disseminated via *Nutrients* (175 articles) and *Food & Function*.

The keyword evolution reflects a paradigm shift: early studies focused on “insulin resistance” (906 occurrences, 2005–2010) and “hepatic steatosis” (892), while the 2012 nomenclature harmonization and the 2015 introduction of NASH histological grading shifted attention toward inflammatory processes [3,34]. Since 2020, “omega-3 fatty acids”, “delta-5 desaturase”, and “*Torreya grandis* oil” have gained prominence, indicating a transition from descriptive pathology to fatty-acid-targeted, evidence-based nutritional interventions.

In this study, the most influential work by Donnelly et al. (2743 citations) established that both circulating NEFAs and hepatic de novo lipogenesis drive lipid accumulation in NAFLD [23]. This study provided the pathogenic framework for the field and established stable isotope tracing as a methodological benchmark.

The second most cited study by Marchesini et al. (1886 citations) first defined NAFLD as a hepatic manifestation of metabolic syndrome, introducing the key concept of “liver-specific insulin resistance” [24], challenging the previously held belief that obesity is a prerequisite for NAFLD and highlighting its presence in non-obese populations with underlying metabolic disturbances.

The third most cited publication, “Obesity and Nonalcoholic Fatty Liver Disease: Biochemical, Metabolic, and Clinical Implications”, by Fabbrini et al. has been cited 1569 times [25]. It consolidated the pathophysiological model by identifying dysfunctional fatty acid metabolism, adipose tissue dysfunction, and hepatic steatosis as the core mechanisms linking NAFLD to systemic cardiometabolic risk [25].

In recent years, the mechanisms through which fatty acids influence the onset and progression of NAFLD have become increasingly understood [35]. Aberrant hepatic lipid accumulation is established as the central pathological hallmark of NAFLD and is closely associated with metabolic dysregulation, oxidative stress, insulin resistance, and chronic inflammation. A study by Rodrigo Valenzuela et al. further demonstrated that unsaturated fatty acids promote fatty acid oxidation via PPAR-α activation and suppress lipogenesis through downregulation of SREBP-1c, while their oxidation products activate Nrf2, eliciting antioxidant and anti-fibrotic responses [36].The evolving landscape of NAFLD research, as revealed by our bibliometric analysis, demonstrates a clear paradigm shift from a focus on foundational pathological mechanisms to a concerted effort in developing nutritional interventions. This shift is not merely a change in topic popularity but represents a maturation of the field. Early research clusters centered on “insulin resistance” and “oxidative stress” provided the critical mechanistic justification for exploring dietary lipids as a therapeutic lever [37]. The subsequent emergence and consolidation of strong clusters around “ω-3 fatty acids” and “double-blind trials” signify a translational phase where these mechanistic insights are being rigorously tested in humans [38]. Furthermore, the recent rise of “short-chain fatty acids” as a keyword burst points to the next frontier: the exploration of the gut–liver axis and the role of the microbiome in NAFLD pathophysiology [39]. Within this context, the role of unsaturated fatty acids in both the prevention and management of NAFLD has attracted considerable scholarly interest.

Our findings provide a data-driven justification for public health action. They also contribute directly to the treatment of NAFLD by offering an evidence-based roadmap for nutritional therapy. The strong co-occurrence of ω-3 PUFAs with clinical trial keywords underscores their position as the most substantiated nutritional supplement for NAFLD management, supporting their inclusion in dietary guidelines [40]. Conversely, our analysis identifies a conspicuous translational gap: despite preclinical promise, other unsaturated fatty acids like ω-6 and ω-9, as well as specific nutraceutical oils such as *Torreya grandis* oil, remain significantly underrepresented in the clinical trial literature [41]

In a randomized, multicenter, double-blind, placebo-controlled trial, Janczyk et al reported that omega-3 fatty acid supplementation significantly improved NAFLD outcomes in pediatric populations [42]. Similarly, a randomized, double-blind, controlled trial by Rezaei et al investigated the effects of flaxseed oil versus sunflower oil in conjunction with a low-fat diet in 68 NAFLD patients over a 12-week period [43]. The participants in the flaxseed oil group showed a significantly lower fatty liver grade and notable reductions in blood glucose and lipid levels, suggesting that flaxseed oil may be more effective in improving metabolic parameters associated with NAFLD [43]. Emerging research has also begun to explore other members of the unsaturated fatty acid family, including delta-5 fatty acids of the omega-6 series and omega-9 fatty acids, as potential therapeutic agents in the context of NAFLD.

This study has several limitations. First, the analysis focused exclusively on high-quality, peer-reviewed literature indexed in the WOS Core Collection and restricted to English-language articles and reviews. As a result, conference abstracts, case reports, and publications in other languages were excluded, which may have led to the omission of novel or emerging research contributions. Secondly, while recently published studies with low citation counts were included, their current citation status may not accurately reflect their future academic impact. This limitation is inherent to the temporal constraints of bibliometric analysis and highlights the evolving nature of scientific influence over time.

## 5. Conclusions

Through bibliometric analysis, this study identified China as a principal contributor in terms of total research output, while the United States demonstrated stronger academic influence and more extensive international collaborations. Our study makes two contributions: firstly, it contributes to the study of the disease by objectively mapping its knowledge evolution and highlighting emerging paradigms like SCFAs and the gut–liver axis. Secondly, and more critically, it contributes to its treatment by translating bibliometric trends into an actionable roadmap. This roadmap confirms the evidence base for ω-3 PUFAs while simultaneously identifying underexplored fatty acids such as ω-6/ω-9 and specific botanical oils as high-priority candidates for future clinical investigation. Bridging these identified gaps is essential for expanding the arsenal of evidence-based, non-pharmacological therapies for this prevalent liver disease.

## Figures and Tables

**Figure 1 foods-14-04277-f001:**
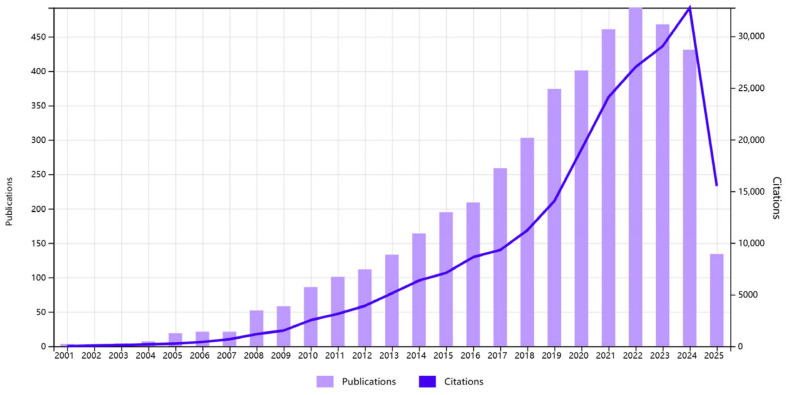
Number of publications and citations per year (data from WOS, until 11 June 2025).

**Figure 2 foods-14-04277-f002:**
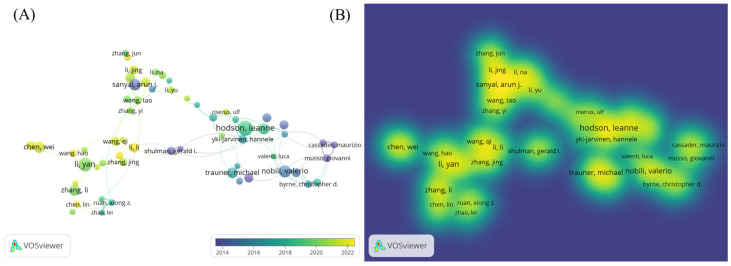
(**A**) Author collaboration network diagram; (**B**) author distribution hotspots.

**Figure 3 foods-14-04277-f003:**
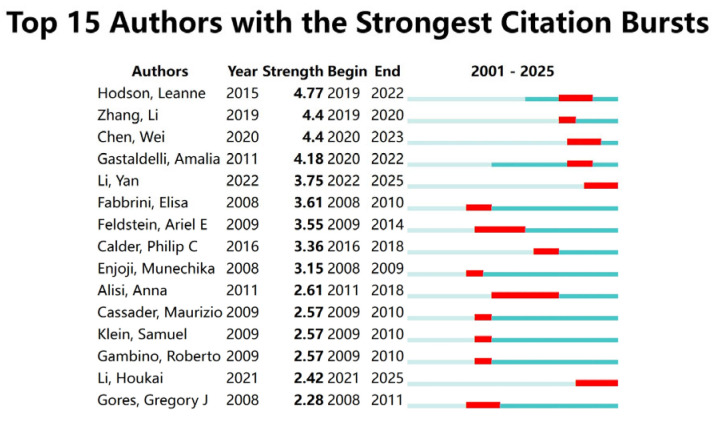
Author explosive ranking.

**Figure 4 foods-14-04277-f004:**
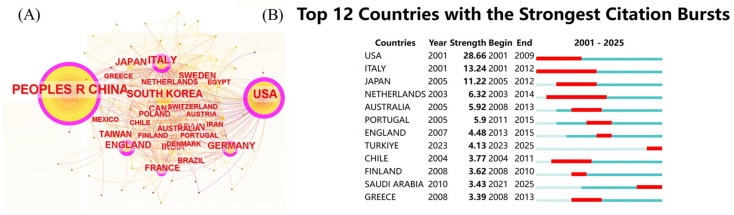
(**A**) National cooperation network diagram; (**B**) national explosive ranking.

**Figure 5 foods-14-04277-f005:**
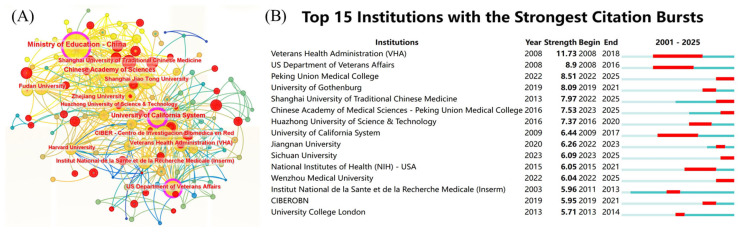
(**A**) Institutional cooperation network diagram; (**B**) institution explosive ranking.

**Figure 6 foods-14-04277-f006:**
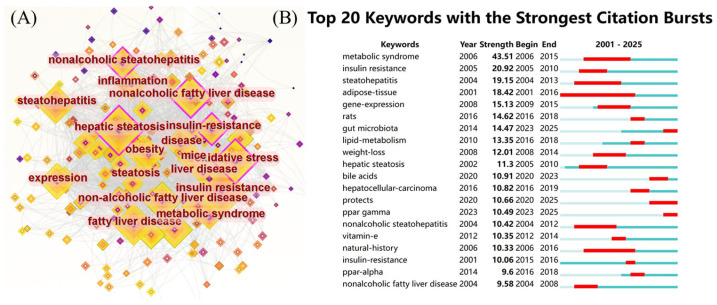
(**A**) Keyword co-occurrence network figure; (**B**) keyword explosive ranking.

**Figure 7 foods-14-04277-f007:**
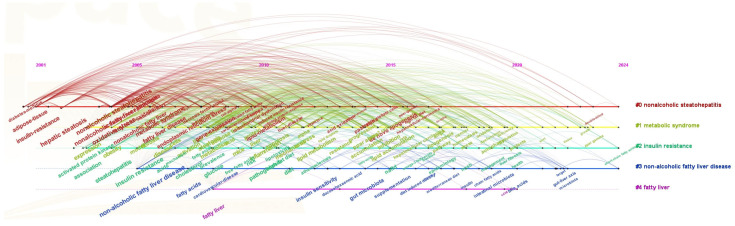
Keyword timeline.

**Table 1 foods-14-04277-t001:** Top 10 authors by number of publications.

Rank	Author	Number of Publications	Affiliated Organization	Total Number of Citations	Average Number of Citations	H-Index
1	Li Y	71	New Drug Research Center, Second Military Medical University	2090	29.44	27
2	Zhang Y	57	Tianjin State Key Laboratory of Modern Chinese Medicine	1452	25.47	19
3	Li J	56	Tianjin State Key Laboratory of Modern Chinese Medicine	1342	23.96	21
4	Zhang J	45	Department of Gastroenterology, the First Affiliated Hospital, College of Medicine, Zhejiang University	1148	25.51	20
5	Wang H	42	Laboratory of Liver Diseases, National Institute on Alcohol Abuse and Alcoholism, National Institutes of Health	1461	34.79	21
6	Liu Y	41	Department of Gastroenterology, Renji Hospital, Shanghai Jiao Tong University School of Medicine, Shanghai Institute of Digestive Disease	1623	39.59	22
7	Wang Y	39	National Laboratory of Biomacromolecules, Institute of Biophysics, Chinese Academy of Sciences, Beijing, China; University of Chinese Academy of Sciences, Beijing, China;	1098	28.15	17
8	Wang X	37	Department of Microbiology and Immunology, Medical College of Virginia and McGuire Veterans Affairs Medical Center, Virginia Commonwealth University	985	26.62	16
9	Wang J	36	Department of Endocrinology and Metabolism, First Affiliated Hospital of Nanchang University	877	24.36	15
10	Chen Y	35	National Collaborative Innovation Center for Diagnosis and Treatment of Infectious Diseases, State Key Laboratory for Diagnosis and Treatment of Infectious Diseases, the First Affiliated Hospital, School of Medicine, Zhejiang University	1304	37.26	16

**Table 2 foods-14-04277-t002:** Top 10 countries based on number of articles.

Rank	Country	Articles	Total Citations	Average Citation	H-Index
1	China	1690	47,648	28.19	93
2	Usa	1031	87,313	84.61	147
3	South Korea	302	9091	30.1	47
4	Italy	300	26,417	88.06	80
5	Japan	242	11,961	49.43	58
6	Germany	190	12,019	63.26	50
7	Spain	182	12,058	66.25	52
8	England	178	11,918	66.96	57
9	Canada	128	7734	60.42	45
10	France	123	9866	80.21	43

**Table 3 foods-14-04277-t003:** Top 10 institutions based on number of articles.

Rank	Institution	Number of Articles	Country	Total Number of Citations	Average Number of Citations	H-Index
1	Ministry Of Education China	166	China	4697	28.3	42
2	Chinese Academy of Sciences	108	China	3700	34.26	36
3	University Of California System	102	USA	9583	93.95	44
4	Ciber Centro De Investigacion Biomedica En Red	98	USA	8392	85.63	43
5	Shanghai Jiao Tong University	97	China	4073	41.99	38
6	Us Department of Veterans Affairs	91	USA	8158	89.65	41
7	Veterans Health Administration (VHA)	90	USA	8145	90.5	41
8	Institut National De La Sante Et De La Recherche Medicale Inserm	82	France	8149	99.38	38
9	Shanghai University of Traditional Chinese Medicine	79	China	1830	23.16	22
10	Zhejiang University	78	China	2780	35.64	31

**Table 4 foods-14-04277-t004:** Top 10 journals based on number of articles.

Rank	Journal	Number of Articles	IF as of 2024	Total Number of Citations	Average Number of Citations	CiteScore Ranking	H-Index
1	*Nutrients*	175	4.8	5732	32.75	Q1	38
2	*International Journal of Molecular Sciences*	135	4.9	3757	27.83	Q1	31
3	*Hepatology*	96	13	16,786	174.85	Q1	64
4	*Scientific Reports*	79	3.8	3524	44.61	Q1	35
5	*Plos One*	67	2.9	3119	46.55	Q1	34
6	*Journal of Hepatology*	65	26.8	11,540	117.54	Q1	50
7	*Journal of Nutritional Biochemistry*	61	4.8	2188	35.87	Q1	26
8	*Food & Function*	59	5.1	1665	128.22	Q1	24
9	*Frontiers In Pharmacology*	54	4.4	1270	23.52	Q1	20
10	*World Journal of Gastroenterology*	50	4.3	3921	78.42	Q1	31

**Table 5 foods-14-04277-t005:** Top 10 publications based on number of citations.

Rank	Total Number of Citations	Studies Name	Author and Year	Journal	IF
1	2743	Sources of fatty acids stored in liver and secreted via lipoproteins in patients with nonalcoholic fatty liver disease	[23]	*Journal of Clinical Investigation*	13.3
2	1886	Nonalcoholic fatty liver disease—A feature of the metabolic syndrome	[24]	*Diabetes*	6.2
3	1569	Obesity and Nonalcoholic Fatty Liver Disease: Biochemical, Metabolic, and Clinical Implications	[25]	*Hepatology*	13
4	1183	Molecular mechanism of PPARα action and its impact on lipid metabolism, inflammation and fibrosis in non-alcoholic fatty liver disease	[26]	*Journal of Hepatology*	26.8
5	1047	A lipidomic analysis of nonalcoholic fatty liver disease	[27]	*Hepatology*	13
6	988	Inflammation in obesity, diabetes, and related disorders	[28]	*Immunity*	25.5
7	988	Molecular mechanisms of hepatic lipid accumulation in non-alcoholic fatty liver disease	[29]	*Cellular And Molecular Life Sciences*	6.2
8	973	Gut microbial metabolites in obesity, NAFLD and T2DM	[30]	*Nature Reviews Endocrinology*	31
9	971	Contribution of de novo fatty acid synthesis to hepatic steatosis and insulin resistance: lessons from genetically engineered mice	[31]	*Journal of Clinical Investigation*	13.3
10	929	Treatment of NAFLD with diet, physical activity and exercise	[32]	*Journal of Hepatology*	26.8

## Data Availability

The data presented in this study are available on request from the corresponding author. The data are not publicly available due to privacy restrictions.

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
