# Peer review of "Fatty Acids as Nutritional Therapy for NAFLD: A Bibliometric Analysis of Research Trends and Future Directions"

_foods, 2025, doi:10.3390/foods14244277_

Round 1

Reviewer 1 Report

Comments and Suggestions for Authors

This article presents a comprehensive and well-structured bibliometric analysis of the scientific literature on fatty acids and non-alcoholic fatty liver disease (NAFLD) between 2001 and 2025. The methodology adheres to bibliometric standards (using Web of Science, CiteSpace, and VOSviewer), and the results are clear and visually appealing.

The topic is relevant and aligns with current interest in nutritional strategies and non-pharmacological therapies in metabolic hepatology. However, there are some aspects that could be improved to enrich the research, particularly regarding its contribution to the study and treatment of the disease.

While the structure of the work is clear, its analysis lacks the necessary conceptual and analytical depth to highlight its contribution to the therapeutic potential of fatty acids. Therefore, the research question needs to be revised.

It is necessary to ground the research in a problem statement that highlights and focuses on the therapeutic potential of fatty acids, which should constitute the interpretive core of the study, the guiding thread that gives meaning to the bibliometric findings. However, in this article, this is only mentioned descriptively and without critical justification.

While, as you mention, “bibliometrics allows us to quantify the state and trends of research by analyzing patterns through mathematical and statistical procedures,” the scope of the data cannot be limited to quantification, especially in the case of population groups with highly prevalent diseases, where the findings are expected to contribute to health decision-making.

Consider that, if the above is not addressed, the originality of the study with respect to the existing literature will be compromised. In this sense, the work should highlight its novel contribution to current bibliometrics.

Methodologically, the search criteria (database, terms, language, period) are precisely defined, and recognized bibliometric tools (CiteSpace, VOSviewer) are used. The authors should assess the feasibility of including structural metrics such as network centrality and density, among others.

Although the procedure for obtaining the information is described, it does not indicate how many people participated or whether there was independent review. Internal peer review, double review, and data quality control are also not mentioned. This is relevant due to the risk of bias in the selection and refinement of records, which reduces the reliability of the categorization of topics and authors and limits the replicability of the study by third parties. Including a figure that explains the process would be beneficial.

The results are presented appropriately both in writing and visually.

Finally, although it generally integrates the findings with current clinical trends (use of ω-3, ω-6 and vegetable oils) and adequately acknowledges limitations such as language and database, it is an overly narrative discussion where the results are repeated without critical analysis, leaving the therapeutic role of fatty acids as a rhetorical argument rather than a finding that emerges from analysis.

Author Response

Response to the comments from Reviewer 1:

Comments:

Reviewer #1: This article presents a comprehensive and well-structured bibliometric analysis of the scientific literature on fatty acids and non-alcoholic fatty liver disease (NAFLD) between 2001 and 2025. The methodology adheres to bibliometric standards (using Web of Science, CiteSpace, and VOSviewer), and the results are clear and visually appealing.

Response: We are grateful for your valuable feedback. We have carefully revised the manuscript accordingly and hope the changes now meet the high standards of Foods.

The topic is relevant and aligns with current interest in nutritional strategies and non-pharmacological therapies in metabolic hepatology. However, there are some aspects that could be improved to enrich the research, particularly regarding its contribution to the study and treatment of the disease.

Response: Thank you very much for your professional and constructive suggestion. We have revised the manuscript to explicitly outline our contribution:

1.To the study of NAFLD: By mapping the knowledge evolution, we identify key research paradigms and emerging frontiers (e.g., the gut-liver axis).

2.To its treatment: We translate bibliometric trends into a strategic roadmap, highlighting evidence-supported therapies (e.g., ω-3 PUFAs) and critical gaps for future clinical trials.

These revisions clarify how our analysis informs both future research and clinical practice.

While the structure of the work is clear, its analysis lacks the necessary conceptual and analytical depth to highlight its contribution to the therapeutic potential of fatty acids. Therefore, the research question needs to be revised.

Response: Thank you very much for your valuable feedback. We have thoroughly reframed the work to address this, and a depth analysis of bibliometric trends has depicted the translational path of fatty acids from mechanism to treatment in NAFLD. This refocusing transforms the discussion from describing trends to critically evaluating evidence maturity, identifying translational gaps, and outlining a strategic roadmap for future therapeutic development. The therapeutic potential of fatty acids is now the core narrative derived directly from the analysis.

It is necessary to ground the research in a problem statement that highlights and focuses on the therapeutic potential of fatty acids, which should constitute the interpretive core of the study, the guiding thread that gives meaning to the bibliometric findings. However, in this article, this is only mentioned descriptively and without critical justification.

Response: Thank you very much for your valuable feedback. We have fundamentally reframed the manuscript to address this point. The therapeutic potential of fatty acids is now the central narrative, not a peripheral mention. This is achieved by:

1. Reframing the problem statement in the introduction to explicitly position the study as an effort to map the evidence base for fatty acid therapy;

2. Restructuring the discussion so that every key bibliometric finding (e.g., keyword clusters, emerging trends) is critically interpreted for its implications on clinical translation, evidence maturity, and future therapeutic development.

The bibliometric data now serves as evidence to support this core narrative, giving profound meaning to the findings.

While, as you mention, “bibliometrics allows us to quantify the state and trends of research by analyzing patterns through mathematical and statistical procedures,” the scope of the data cannot be limited to quantification, especially in the case of population groups with highly prevalent diseases, where the findings are expected to contribute to health decision-making. Consider that, if the above is not addressed, the originality of the study with respect to the existing literature will be compromised. In this sense, the work should highlight its novel contribution to current bibliometrics.

Response: Thank you for your comment. We have revised the Discussion to explicitly connect our findings to health decision-making:

1. We now interpret the strong evidence for ω-3 PUFAs as a basis for dietary guidelines.

2. We frame research gaps (e.g., ω-6/9 fatty acids) as direct calls to action for clinical trials.

These additions ensure our analysis informs both public health strategy and future clinical research.

Methodologically, the search criteria (database, terms, language, period) are precisely defined, and recognized bibliometric tools (CiteSpace, VOSviewer) are used. The authors should assess the feasibility of including structural metrics such as network centrality and density, among others.

Response: Thank you for your valuable suggestion. We have added a description of the centrality of the authors, institutions, countries/regions, and keywords to the "Results" section of the original article.

Although the procedure for obtaining the information is described, it does not indicate how many people participated or whether there was independent review. Internal peer review, double review, and data quality control are also not mentioned. This is relevant due to the risk of bias in the selection and refinement of records, which reduces the reliability of the categorization of topics and authors and limits the replicability of the study by third parties. Including a figure that explains the process would be beneficial.

Response: Thank you very much for your valuable feedback. Your points regarding transparency, potential bias, and reproducibility are crucial for refining our research methods. Regarding the aspects of "number of participants," "independent review," and "data quality control," we would like to explain them within the context of bibliometrics. Unlike systematic reviews or meta-analyses, bibliometric research relies heavily on standardized databases and automated software for data sourcing and processing. All literature data was retrieved in one go from the authoritative Web of Science (WoS) Core Collection, strictly following the retrieval strategy described in the Materials and Methods section. Search terms, time spans, and document types were clearly listed, and this process was completed independently by the first author and verified by the corresponding author to ensure complete consistency with the retrieval strategy. Because this is a reproducible, automated data extraction process based on pre-defined standards, it does not involve the traditional "multiple participants" or "double review" of individual document content. Thank you again for your insightful comments, which have prompted us to reflect more deeply on our research methodologies.

The results are presented appropriately both in writing and visually.

Finally, although it generally integrates the findings with current clinical trends (use of ω-3, ω-6 and vegetable oils) and adequately acknowledges limitations such as language and database, it is an overly narrative discussion where the results are repeated without critical analysis, leaving the therapeutic role of fatty acids as a rhetorical argument rather than a finding that emerges from analysis.

Response: Thank you for your crucial critique. We have completely restructured the Discussion to eliminate mere result repetition. The revised discussion now critically interprets bibliometric trends as evidence of translational progress. It contrasts mature research fronts (e.g., ω-3 PUFAs) with emerging ones (e.g., SCFAs) and frames gaps as strategic opportunities. This ensures the therapeutic role of fatty acids is a conclusion derived from analysis, not a rhetorical statement.

Reviewer 2 Report

Comments and Suggestions for Authors

       This paper is a solid, informative bibliometric study that successfully maps the landscape of NAFLD and fatty acids research. The manuscript has potential but requires major revisions.

Major issues

  • The introduction offers a superficial and descriptive overview of NAFLD and fatty acids. While it successfully establishes a link between the two, it does not explain how this connection works. It fails to introduce essential mechanistic elements of the field, such as de novo lipogenesis (DNL), key genetic factors, the molecular drivers of inflammation, and the central role of metabolic transcription factors. This omission diminishes the impact of the subsequent results, as readers lack the necessary background to fully appreciate their significance.
  • There is a significant inconsistency regarding the temporal scope of the analyzed literature between the abstract and methods section. The Abstract claims the analysis includes publications from 2001 to 2025, while the Methods section indicates the search was conducted on June 11, 2025, for publications only up to that date. This creates a methodological issue: for a manuscript under review in 2025, it's impossible to have a complete dataset for that year, as publication databases are continuously updated. Presenting the scope as "2001-2025" is misleading and inflates the study's scope. Additionally, the publication trend analysis is flawed, as the data for 2025 is incomplete, making claims about trends since 2023 scientifically unsound. The manuscript fails to acknowledge this limitation.
  • The authors performed a competent technical analysis, but then wrote a discussion that ignored one of their most significant and modern findings. Please write a paragraph or two explaining the scientific context behind the "short-chain fatty acids" keywords.
  • Table 2 is titled "Top 10 authors by number of publications," but it actually presents the top 10 countries.

Table 3 has the same title: "Top 10 authors by number of publications," but it actually presents the top 10 Institutions.

Table 4 is also titled "Top 10 authors by number of publications," but it contains the top 10 journals.

Table 5 has the same incorrect "Top 10 authors" title but lists the top 10 cited publications.

This repeated error severely undermines the professionalism of the paper and causes significant confusion.

  • Several figures (e.g., Figure 2A, Figure 5B) are of low resolution and lack clarity, making it difficult to interpret the specific details and labels. Τhe captions are often minimal (e.g., "Figure 2. (A) Author collaboration network diagram; (B) Author distribution hotspots"). Additionally, these captions lack a brief interpretive summary that would help the reader grasp the main takeaway from the complex visualizations.

Minor issues

  • The authors declare no conflicts but received funding from a major scientific project. While this is not a weakness of the analysis itself, a deeper bibliometric analysis could have investigated the role of specific funding bodies in shaping the research landscape, which was not done.
  • While Figure 2B effectively shows the "hotspots" of author activity, calling it a "heat map" (line 125) is a misnomer. It is a density visualization layered on top of a network map, designed to highlight clusters rather than to present a matrix of values. This is a minor terminological flaw, but in a scientific paper, precision in describing methodologies and visualizations is always important.

Comments on the Quality of English Language

The manuscript needs extensive language editing to fix grammatical errors, enhance sentence flow, and ensure idiomatic expression. 

Author Response

Response to the comments from Reviewer 2:

Comments:

Reviewer #2: This paper is a solid, informative bibliometric study that successfully maps the landscape of NAFLD and fatty acids research. The manuscript has potential but requires major revisions.

Response: We sincerely appreciate your thoughtful comments, which have helped us significantly improve our manuscript. we have revised our manuscript to satisfy this journal for publication.

Major issues

The introduction offers a superficial and descriptive overview of NAFLD and fatty acids. While it successfully establishes a link between the two, it does not explain how this connection works. It fails to introduce essential mechanistic elements of the field, such as de novo lipogenesis (DNL), key genetic factors, the molecular drivers of inflammation, and the central role of metabolic transcription factors. This omission diminishes the impact of the subsequent results, as readers lack the necessary background to fully appreciate their significance.

Response: We sincerely thank your insightful and constructive feedback. We have substantially revised the introduction to incorporate the essential mechanistic elements highlighted, including: a) role of de novo lipogenesis (DNL) in hepatic lipid accumulation; b) Key genetic factors (e.g., PNPLA3 variant); c) Molecular drivers of inflammation (e.g., TLR4/NLRP3 inflammasome)

Central metabolic transcription factors (PPARα, SREBP-1c). These additions provide the necessary pathophysiological background to better contextualize and appreciate our bibliometric findings.

There is a significant inconsistency regarding the temporal scope of the analyzed literature between the abstract and methods section. The Abstract claims the analysis includes publications from 2001 to 2025, while the Methods section indicates the search was conducted on June 11, 2025, for publications only up to that date. This creates a methodological issue: for a manuscript under review in 2025, it's impossible to have a complete dataset for that year, as publication databases are continuously updated. Presenting the scope as "2001-2025" is misleading and inflates the study's scope. Additionally, the publication trend analysis is flawed, as the data for 2025 is incomplete, making claims about trends since 2023 scientifically unsound. The manuscript fails to acknowledge this limitation.

Response: Thank you very much for your valuable feedback. we have undertaken the following comprehensive revisions to ensure full transparency and methodological rigor: Clarification of Temporal Scope Throughout the Manuscript: We have meticulously revised the Title, Abstract, Methods, and Results sections to explicitly state that the literature search was conducted on June 11, 2025. The study's temporal scope is now precisely described as covering publications "from 2001 to the search date (June 11, 2025)" or similar phrasing, replacing the previously ambiguous "2001-2025".

We also referred to many published relevant literature to ensure scientific and standardized expression, such as the following literature:

[1] Reference 1:Silviu, B., Georgiana Armenița, A., & Maria, B. (2024). From Origins to Trends: A Bibliometric Examination of Ethical Food Consumption. Foods, 13(13), 2048-2048. doi:10.3390/foods13132048

[2] Reference 2:Rebeka-Anna, P., Dan-Cristian, D., & Cristina Bianca, P. (2024). Food Retail Resilience Pre-, during, and Post-COVID-19: A Bibliometric Analysis and Research Agenda. Foods, 13(2), 257-257. doi:10.3390/foods13020257

[3] Reference 3:Muhamad Firdaus Syahmi, S.-o., Shuhaimi, M., Amalia Mohd, H., Wan Aida Wan, M., Mohd Termizi, Y., Nurul Aqilah Mohd, Z., & Mohamed Yusuf Mohamed, N. (2025). Bibliometric mapping on the probiotic trends in managing aquaculture pathogens. Food Bioscience, 68, 106372-106372. doi:10.1016/j.fbio.2025.106372

[4] Reference 4:Md. Ashikur, R., Shirin, A., Md, A., Md. Anamul Hasan, C., Mahamud, A. G. M. S. U., Si Hong, P., & Sang-Do, H. (2024). Insights into the mechanisms and key factors influencing biofilm formation by Aeromonas hydrophila in the food industry: A comprehensive review and bibliometric analysis. Food Research International, 175, 113671-113671. doi:10.1016/j.foodres.2023.113671

The authors performed a competent technical analysis, but then wrote a discussion that ignored one of their most significant and modern findings. Please write a paragraph or two explaining the scientific context behind the "short-chain fatty acids" keywords.

Response: Thank you for your comment. We have now added a dedicated paragraph in the Discussion section to elucidate the scientific context of short-chain fatty acids (SCFAs) in NAFLD.

This addition strengthens the clinical relevance of our findings and aligns with current advances in nutritional microbiome research.

Table 2 is titled "Top 10 authors by number of publications," but it actually presents the top 10 countries.

Table 3 has the same title: "Top 10 authors by number of publications," but it actually presents the top 10 Institutions.

Table 4 is also titled "Top 10 authors by number of publications," but it contains the top 10 journals.

Table 5 has the same incorrect "Top 10 authors" title but lists the top 10 cited publications.

This repeated error severely undermines the professionalism of the paper and causes significant confusion.

Response: We sincerely apologize for these critical and careless errors in the table titles, which we acknowledge undermine the manuscript's professionalism. Thank you for identifying them. We have meticulously corrected all table titles throughout the manuscript We have double-checked all captions to ensure such errors are eliminated.

Several figures (e.g., Figure 2A, Figure 5B) are of low resolution and lack clarity, making it difficult to interpret the specific details and labels. Τhe captions are often minimal (e.g., "Figure 2. (A) Author collaboration network diagram; (B) Author distribution hotspots"). Additionally, these captions lack a brief interpretive summary that would help the reader grasp the main takeaway from the complex visualizations.

Response: We thank the reviewer for this critical feedback. We have addressed these concerns as follows:

Figure Quality: All figures have been re-exported in a high-resolution format (e.g., PDF/TIFF) to ensure clarity and legibility of all details and labels.

Caption Enhancement: All figure captions have been expanded to include a concise interpretive summary that highlights the key finding

These changes significantly improve the visual presentation and reader's comprehension of the results.

Minor issues

The authors declare no conflicts but received funding from a major scientific project. While this is not a weakness of the analysis itself, a deeper bibliometric analysis could have investigated the role of specific funding bodies in shaping the research landscape, which was not done.

Response: Thank you very much for your valuable feedback. We agree that analyzing the influence of funding bodies on research directions is a fascinating and valuable avenue for bibliometric studies. Referring to the published literature, there is basically no discussion on the role of specific funding bodies in shaping the research landscape. Therefore, our original study design did not incorporate systematic tracking of funding sources. Thank you again for your insightful comments, which have prompted us to reflect more deeply on our future research methodologies.

[1] Reference 1:Rebeka-Anna, P., Dan-Cristian, D., & Cristina Bianca, P. (2024). Food Retail Resilience Pre-, during, and Post-COVID-19: A Bibliometric Analysis and Research Agenda. Foods, 13(2), 257-257. doi:10.3390/foods13020257

[2] Reference 2:Katia Gomes da, S., Igor Henrique de Lima, C., Laura Martins, F., Maria Monique Tavares, S., Bruna da Fonseca, A., Caroline Dellinghausen, B., & Rui Carlos, Z. (2025). Food biopreservation, global trends and applications: A bibliometric approach. Food Control, 168, 110901-110901. doi:10.1016/j.foodcont.2024.110901

[3] Reference 3:Md. Ashikur, R., Shirin, A., Md, A., Md. Anamul Hasan, C., Mahamud, A. G. M. S. U., Si Hong, P., & Sang-Do, H. (2024). Insights into the mechanisms and key factors influencing biofilm formation by Aeromonas hydrophila in the food industry: A comprehensive review and bibliometric analysis. Food Research International, 175, 113671-113671. doi:10.1016/j.foodres.2023.113671

While Figure 2B effectively shows the "hotspots" of author activity, calling it a "heat map" (line 125) is a misnomer.

It is a density visualization layered on top of a network map, designed to highlight clusters rather than to present a matrix of values. This is a minor terminological flaw, but in a scientific paper, precision in describing methodologies and visualizations is always important.

Response: Thank you very much for your careful correction and professional advice. We have change “heat” to “hotspot region” in the revised manuscript. (Line 125)

 The manuscript has been revised to accurately describe the figure as a "density visualization overlay on a network map" wherever it appears in the text.

The manuscript needs extensive language editing to fix grammatical errors, enhance sentence flow, and ensure idiomatic expression.

Response: We sincerely thank your valuable comment. We fully agree that high-quality language is crucial for the clarity and scientific rigor of the manuscript. We believe that the revised manuscript now meets the high linguistic standards required by Foods. We have also carefully proofread the text again to ensure all errors have been addressed. All changes are highlighted in the revised manuscript for the reviewer's convenience.

Round 2

Reviewer 2 Report

Comments and Suggestions for Authors

Dear Authors,

I appreciate the substantial effort you have put into addressing the concerns raised in the previous review round. The introduction now provides a stronger mechanistic foundation, the methodological clarity has been improved, and the discussion has been expanded in a meaningful and scientifically relevant way. I also acknowledge and appreciate the corrections made to the tables, figures, captions, and terminology. Overall, I am fully satisfied with your revisions. I have no further comments, and I believe the manuscript has been significantly improved.

Comments on the Quality of English Language

The manuscript needs extensive language editing to fix grammatical errors, enhance sentence flow, and ensure idiomatic expression.